# Septin/anillin filaments scaffold central nervous system myelin to accelerate nerve conduction

**Julia Patzig[1], Michelle S Erwig[1], Stefan Tenzer[2], Kathrin Kusch[1], Payam Dibaj[1], Wiebke Möbius[1,3], Sandra Goebbels[1], Nicole Schaeren-Wiemers[4], Klaus-Armin Nave[1,3], Hauke B Werner[1]\***

[1]Department of Neurogenetics, Max Planck Institute of Experimental Medicine, Goettingen, Germany; [2]Institute of Immunology, University Medical Center, Johannes Gutenberg University, Mainz, Germany; [3]Center for Nanoscale Microscopy and Molecular Physiology of the Brain, Göttingen, Germany; [4]Departement of Biomedicine, University Hospital Basel, Basel, Switzerland

**Abstract** Myelination of axons facilitates rapid impulse propagation in the nervous system. The axon/myelin-unit becomes impaired in myelin-related disorders and upon normal aging. However, the molecular cause of many pathological features, including the frequently observed myelin outfoldings, remained unknown. Using label-free quantitative proteomics, we find that the presence of myelin outfoldings correlates with a loss of cytoskeletal septins in myelin. Regulated by phosphatidylinositol-(4,5)-bisphosphate (PI(4,5)P$_2$)-levels, myelin septins (SEPT2/SEPT4/SEPT7/SEPT8) and the PI(4,5)P$_2$-adaptor anillin form previously unrecognized filaments that extend longitudinally along myelinated axons. By confocal microscopy and immunogold-electron microscopy, these filaments are localized to the non-compacted adaxonal myelin compartment. Genetic disruption of these filaments in *Sept8*-mutant mice causes myelin outfoldings as a very specific neuropathology. Septin filaments thus serve an important function in scaffolding the axon/myelin-unit, evidently a late stage of myelin maturation. We propose that pathological or aging-associated diminishment of the septin/anillin-scaffold causes myelin outfoldings that impair the normal nerve conduction velocity.

\*For correspondence: hauke@em.mpg.de

**Competing interests:** The authors declare that no competing interests exist.

## Introduction

Fast nerve conduction is crucial for normal motor, sensory, and cognitive abilities. In vertebrates, rapid ('saltatory') signal propagation is achieved by the myelination of axons. Myelination is largely completed before adulthood, notwithstanding that adult-born oligodendrocytes can assemble new myelin sheaths throughout life (*Nave and Werner, 2014*). Yet, myelin is one of the most long-lived structures in the CNS (*Toyama et al., 2013*). With age however, the abundance and dynamics of CNS myelin decreases (*Lasiene et al., 2009*) while there is an increase in the frequency of pathological redundant myelin sheaths (*Peters, 2002*; *Sturrock, 1976*), i.e., local outfoldings of compact myelin with normal-appearing axo-myelinic interface. These myelin outfoldings emerge from the innermost, adaxonal myelin layer, a part of the non-compacted, cytoplasmic channel system through myelin. This compartment of myelin has previously been implicated in glia-axonal metabolic coupling (*Nave and Werner, 2014*). Analogous focal outfoldings of peripheral myelin are the pathological hallmark of tomaculous neuropathies affecting the PNS. The normal structure of myelin thus requires stabilization, which can fail upon normal aging and in myelin-related disorders. However, the molecular mechanisms that might prevent myelin outfoldings have remained unknown.

**eLife digest** Normal communication within the brain or between the brain and other parts of the body requires information to flow quickly around the nervous system. This information travels along nerve cells in the form of electrical signals. To speed up the signals, a part of the nerve cell called the axon is frequently wrapped in an electrically insulating sheath made up of a membrane structure called myelin.

The myelin sheath becomes impaired as a result of disease or ageing. In order to understand what might produce these changes, Patzig et al. have used biochemical and microscopy techniques to study mice that had similar defects in their myelin sheaths.

The study reveals that forming a normal myelin sheath around an axon requires a newly identified 'scaffold' made of a group of proteins called the septins. Combining with another protein called anillin, septins assemble into filaments in the myelin sheath. These filaments then knit together into a scaffold that grows lengthways along the myelin-wrapped axon. Without this scaffold, the myelin sheath grew defects known as outfoldings. Axons transmitted electrical signals much more slowly than normal when the septin scaffold was missing from the myelin sheath.

Future studies are needed to understand the factors that control how the septin scaffold forms. This could help to reveal ways of reversing the changes that alter the myelin sheath during ageing and disease.

A striking variety of myelin-related genes causes – when mutated in human disorders and in animal models - common pathological features including axonopathy, neuroinflammation, hypomyelination, and structural impairments affecting myelin. For example, abundant constituents of CNS myelin such as proteolipid protein (PLP), cyclic nucleotide phosphodiesterase (CNP), and myelin associated glycoprotein (MAG), are not essential for the biogenesis of myelin per se but their deficiency in mice causes complex CNS pathology (*Edgar et al., 2009*; *Griffiths et al., 1998*; *Li et al., 1994*; *Montag et al., 1994*). The neuropathological profiles observed in these and other myelin mutants are highly overlapping, which has made it difficult to explain distinct aspects of neuropathology by the loss of individual structural myelin proteins. Here, we followed the hypothesis that mutations of single genes can have secondary consequences for the entire protein composition of myelin, which allow elucidating the molecular cause of distinct neuropathological features.

We report the discovery of a previously unrecognized filamentous scaffold in the innermost layer of CNS myelin that extends longitudinally along myelinated axons. This filament is composed of distinct septin monomers (SEPT2/SEPT4/SEPT7/SEPT8) and associated with the adaptor protein anillin (ANLN). The formation of myelin septin filaments is a late stage of myelin maturation, thereby avoiding that the property of septins to rigidify the membranes they are associated with (*Gilden and Krummel, 2010*; *Tooley et al., 2009*) could hinder the normal developmental ensheathment of axons. Importantly, this sub-membranous septin/anillin-scaffold (SAS) is required for the normal structure of the axon/myelin unit in vivo as its deficiency causes a specific structural impairment of the myelin sheath, myelin outfoldings, and reduced nerve conduction velocity. These findings were possible by systematic label-free myelin proteome analysis in several models of complex neuropathology.

## Results

### Neuropathology and myelin proteome analysis in myelin mutants

By quantitative evaluation of electron micrographs (*Figure 1A*; *Figure 1—figure supplement 1A–E*), hypomyelination was a significant feature in $Plp1^{null}$ and $Mag^{null}$ mice, swellings of the inner tongue of myelin in $Mag^{null}$ and $Cnp^{null}$ mice, axonal spheroids in $Plp1^{null}$ mice, and degenerated axons in $Cnp^{null}$ mice. Split myelin lamellae (not quantified) were observed in $Plp1^{null}$ mice. Together, these mouse mutants display distinct but overlapping profiles of neuropathology. Notably, myelin outfoldings (*Figure 1B*; *Figure 1—figure supplement 1A*) were common to all three models, suggesting a common molecular mechanism.

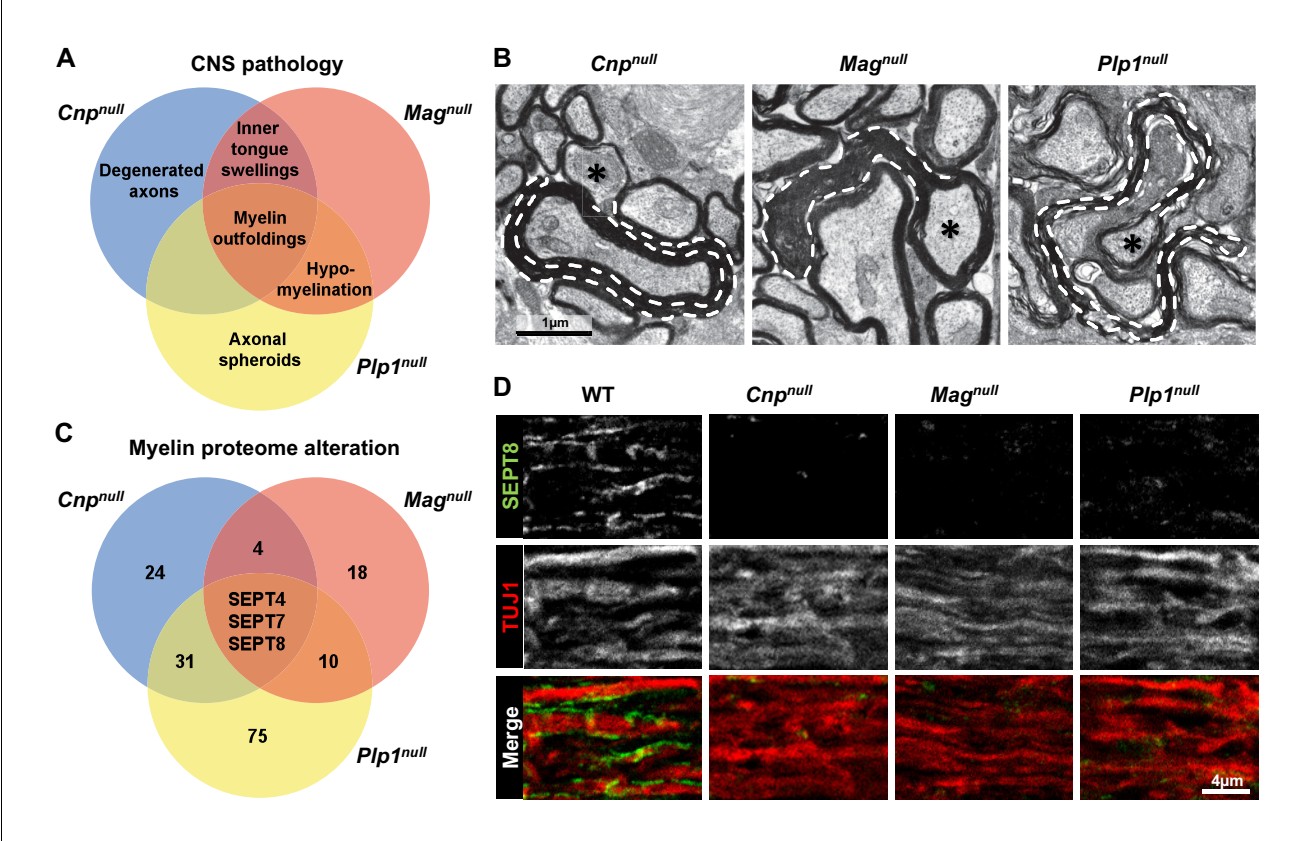

**Figure 1.** Mouse mutants with complex CNS pathology exhibit distinct but overlapping changes of myelin composition. (A) Venn diagram summarizing CNS pathology in mice lacking the myelin proteins CNP, MAG, or PLP, according to quantitative evaluation of electron micrographs of optic nerves at P75. Note that myelin outfoldings are common to all analyzed mutants. See *Figure 1—figure supplement 1A–E* for quantification of these experiments. (B) Electron micrographs of P75 optic nerve cross-sections showing several myelinated axons. Myelin outfoldings and associated axons are labelled with stippled lines and asterisks, respectively. Images are representative of 4 animals per genotype, as quantified in *Figure 1—figure supplement 1A*. (C) Venn diagram summarizing myelin proteome alterations in *Cnp*[null], *Mag*[null], and *Plp1*[null]-mice determined by quantitative mass spectrometry. Given are the numbers of proteins exhibiting significantly changed abundance in myelin purified from the brains of respective mutants at P75. Note that several septins are diminished in all analyzed mutants. n=3 animals per genotype. See *Figure 1—source data 1* for entire dataset and *Figure 1—figure supplement 1G–I* for validation by immunoblot. (D) Immunolabelling validates diminishment of SEPT8 (green) in myelinated fibre tracts of *Cnp*[null], *Mag*[null], and *Plp1*[null]-mice. Longitudinally sectioned optic nerves of P75 mice are shown. The axonal marker TUJ1 (red) was co-labelled as a control. Images are representative of three independent experiments.

The following source data and figure supplement are available for figure 1:

**Source data 1.** Dataset file (differential myelin proteome analyses) related to *Figure 1C*, *Figure 1—figure supplement 1F*, and *Figure 6E*.

**Figure supplement 1.** Neuropathological and molecular analysis in mouse models of complex CNS pathology (*Cnp*[null], *Mag*[null], and *Plp1*[null] mice).

By subjecting myelin purified from the brains of these models to label-free quantitative mass spectrometry (*Distler et al., 2014*), we found distinct but overlapping alterations of the myelin proteome (*Figure 1C*; *Figure 1—source data 1*). Notably, the abundance of several septins was reduced in all analyzed mutants (*Figure 1C*). Septins have important functions in physiology and cell division (*Dolat et al., 2014*; *Fung et al., 2014*), but their role in myelinating cells is unknown (*Buser et al., 2009*; *Patzig et al., 2014*). By mass spectrometry (*Figure 1—source data 1*), the most abundant septins in wild-type CNS myelin are SEPT2, SEPT4, SEPT7, and SEPT8. The abundance of all four septins was reduced in myelin of all three mutants (*Figure 1—figure supplement 1F*), as validated by immunoblotting (*Figure 1—figure supplement 1G–I*). Immunohistochemical analysis of optic nerves confirmed the diminishment of SEPT8 in all three mutants (*Figure 1D*). By qRT-PCR, we

could amplify cDNA fragments for *Sept2*, *Sept4*, *Sept7*, and *Sept8* from mutant and control corpus callosi with approximately similar efficiency, suggesting that transcriptional regulation is unlikely to cause the loss of myelin septins (*Figure 1—figure supplement 1J–M*). Together, the presence of myelin outfoldings correlates with loss of myelin septins in three models of complex CNS pathology.

## Identification of a septin filament in the adaxonal compartment of myelin

Forming membrane-associated hetero-oligomers and higher-order structures (*Bridges et al., 2014*; *Sirajuddin et al., 2007*), septins can rigidify plasma membranes (*Gilden and Krummel, 2010*). By mass spectrometry (*Figure 1—source data 1*), SEPT2, SEPT4, SEPT7, and SEPT8 have a molar stoichiometry of about 1:1:2:2 in normal myelin. SEPT9 was also detected, but at much lower abundance. Thus, myelin comprises septin subunits from all four homology groups, a likely prerequisite for their assembly (*Dolat et al., 2014*; *Fung et al., 2014*; *Sirajuddin et al., 2007*).

To determine the localization and higher-order structure of myelin septins, we performed immunohistochemistry and confocal microscopy of longitudinal sections of optic nerves and spinal cords. SEPT7 and SEPT8-labelling was found to parallel (but not overlap with) axonal neurofilament labelling (*Figure 2A–C*, *Video 1*), suggesting the presence of longitudinal septin filaments in myelin. We therefore colabelled SEPT8 and a marker for adaxonal myelin (MAG). In cross sections, SEPT8-immunopositive puncta appeared contained within the ring-shaped compartment defined by MAG-immunopositivity (*Figure 2D*). Any ring-shaped axon/myelin-unit identified by MAG-labelling exhibited between 0–3 puncta of SEPT8-labeling (*Figure 2E*) independent of the axonal diameter (*Figure 2F*). Aiming to reveal the exact localization of SEPT8 in the adaxonal cytoplasmic (i.e. non-compacted) compartment of myelin at higher resolution, we used cryo-immuno electron microscopy of optic nerves. Immunogold labelling of SEPT7 and SEPT8 supported the localization in adaxonal myelin (*Figure 2G–H*, *Figure 2—figure supplement 1A–B*). Interestingly, within this compartment SEPT8 immunogold was mostly associated with the innermost membrane of compact myelin (*Figure 2H′–I*), in difference to MAG, a transmembrane protein localized to the adaxonal membrane.

Taken together, we have identified a previously unrecognized network of longitudinal septin filaments in the adaxonal non-compact layer of CNS myelin (*Figure 3A*). These filaments extend straight or slightly undulating for various lengths up to 20 μm. Septin filaments extend beyond the internodal segments of myelinated axons into the juxtaparanodal regions, as evidenced by occasional proximate-labelling of SEPT8 with Kv1.2 (*Figure 2—figure supplement 1C*). Proximate-labelling with the paranodal marker CASPR was not found (*Figure 2—figure supplement 1D*).

## Septins in myelin maturation and aging

During oligodendroglial maturation, the abundance of *Sept8*-mRNA increases, but later than most myelin markers (*de Monasterio-Schrader et al., 2012*). When we immunoblotted myelin (purified from wild-type mouse brains at P10, P15, P21, P28, representing ongoing myelin maturation), the abundance of SEPT4, SEPT7, and SEPT8 increased with age whereas the abundance of the classical myelin markers PLP and myelin basic protein (MBP) did not change (*Figure 3B*). When co-immunolabelling SEPT8 and MAG in optic nerves (ages P15, P21, P28), MAG was readily detectable at all examined ages (*Figure 3C*). Conversely, SEPT8 was detectable at P28 but not at P15 or P21 (*Figure 3C*). Thus, the assembly of myelin septins represents a late stage of myelin maturation coinciding with a developmentally decreased frequency of myelin outfoldings (*Figure 3D*). Considering the increase of myelin outfoldings in the aging brain ([*Peters, 2002*; *Sturrock, 1976*] and *Figure 3D*), we also assessed the abundance of septins (SEPT2, SEPT4, SEPT7, SEPT8) in myelin purified from wild-type mouse brains at two years of age. Interestingly, their abundance was decreased compared to myelin from young adult brains (*Figure 3E*), though not as strongly as in myelin of *Mag*[null], *Cnp*[null] and *Plp1*[null] mice. Together, the developmental assembly of myelin septins represents a late stage of myelin maturation correlating with a reduced frequency of myelin outfoldings. On the other hand, the increase of myelin outfoldings with normal aging coincides with a reduced abundance of myelin septins.

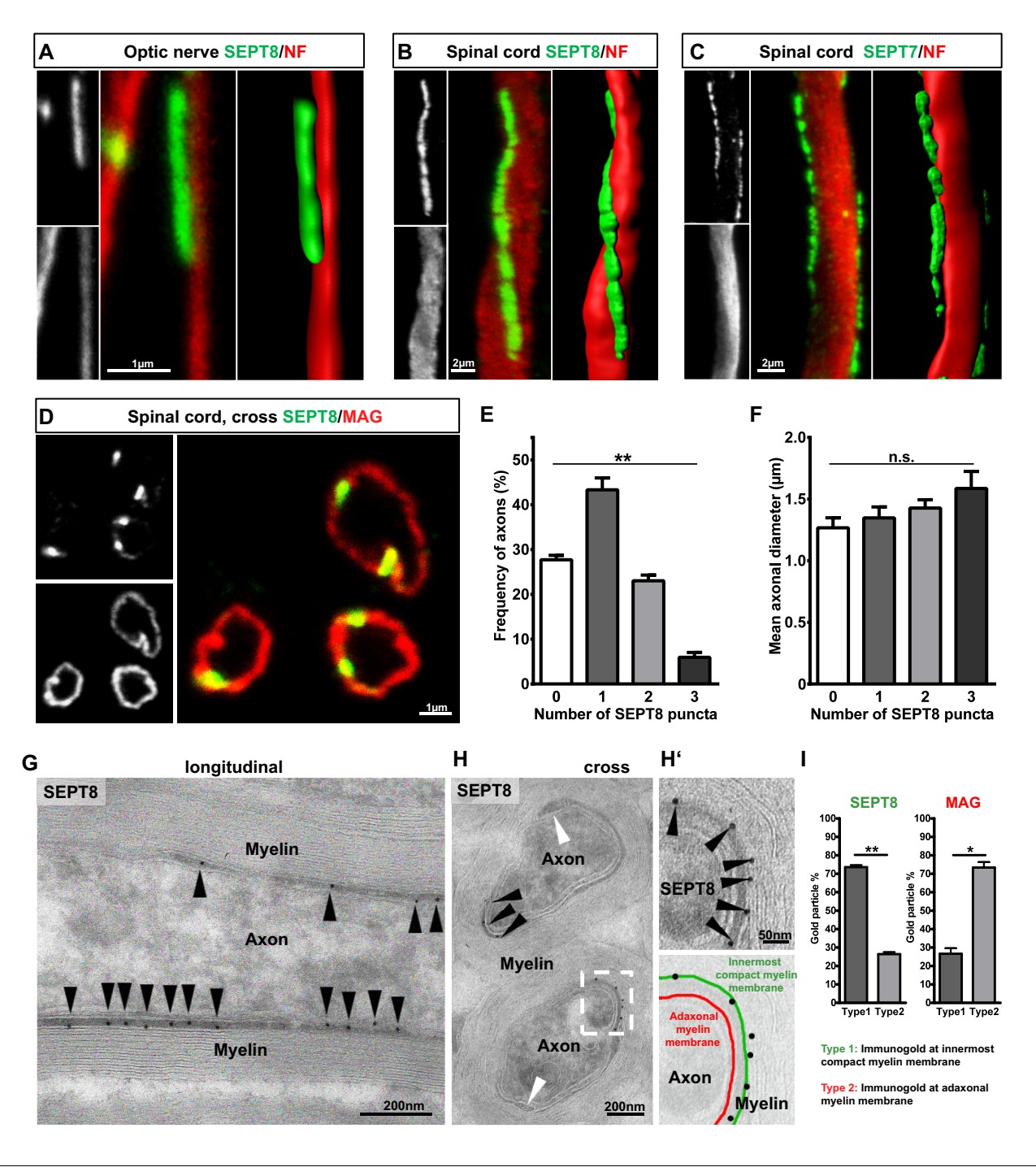

**Figure 2.** Septins form longitudinal filaments in the adaxonal compartment of mature CNS myelin. (A–C) Immunofluorescent signal of SEPT8 and SEPT7 (green) extends longitudinally alongside axons identified by neurofilament-labelling (red). All panels show maximal projections of confocal stacks, and 3-dimensional reconstructions of longitudinally sectioned WT optic nerve (A) or spinal cord (B,C) at age P75. Images are representative of three animals. (D–F) SEPT8 (green) immunolabelling indicates that septin filaments localize to the adaxonal non-compact myelin compartment marked by MAG-immunolabelling (red) (confocal micrograph, D). The number of filaments represented by SEPT8-puncta is plotted in relation to their frequency per axon/myelin-unit (E) and the axonal diameter (F). Data are represented as mean ± SEM. Analysis of 4 animals; repeated-measures-ANOVA; **p=0.0014 (E), n.s., p=0.26 (F). (G,H) Immunogold-labelling of cryosections identifies the localization of SEPT8 in the adaxonal myelin compartment in longitudinally (G) and cross-sectioned (H) optic nerves. Black arrowheads point at immunogold; white arrowhead points at the inner mesaxon. Images are representative of three animals. (H') Enlargement of the boxed area in H shows immunogold labelling of SEPT8 associated with the innermost membrane layer of compact myelin (green false colour in the overlay), not with the adaxonal myelin membrane (red false colour in the overlay). (I)

*Figure 2 continued on next page*

*Figure 2 continued*

Quantification of immunogold labelling of SEPT8 and MAG relative to the innermost membrane layer of compact myelin (type 1) and the adaxonal myelin membrane (type 2). Note that SEPT8 immunogold labelling is associated with the innermost membrane layer of compact myelin while MAG labelling is associated with the adaxonal myelin membrane. Mean ± SEM. Analysis of 3 animals; two-tailed paired t-test; SEPT8 **p=0.002, MAG, *p=0.02.

The following figure supplement is available for figure 2:

**Figure supplement 1.** Localization of myelin septins.

## Association of myelin septins with anillin

The mRNA abundance profile of *Sept8* during oligodendroglial maturation coincides with that of *Anln* (*de Monasterio-Schrader et al., 2012*), which encodes the pleckstrin homology (PH) domain-containing protein anillin (ANLN) (*Menon et al., 2014*; *Oegema et al., 2000*) that can serve as an adaptor to recruit septins onto membranes, at least in yeast (*Liu et al., 2012*). We have thus tested whether anillin is also associated with septins in myelin. When co-immunolabelling SEPT8 and ANLN in the optic nerves of mature wild-type mice, ANLN was readily detectable and co-localized with SEPT8 (*Figure 4A*). Importantly, ANLN immunolabelling was strongly reduced in $Mag^{null}$, $Cnp^{null}$ and $Plp1^{null}$ mice, similar to that of SEPT8 (*Figure 4B*). The diminishment of ANLN was confirmed by immunoblotting of myelin purified from the brains of these mutants (*Figure 4C*). Together, these data imply that myelin septin filaments are associated with anillin.

## A PI(4,5)P$_2$–dependent mechanism of myelin septin assembly

In yeast, the cytoplasmic headgroups of the membrane lipid phosphatidylinositol (4,5)-bisphosphate (PI(4,5)P$_2$) recruit ANLN to the cleavage furrow (*Liu et al., 2012*), thereby mediating the submembraneous polymerization of septins (*Bertin et al., 2010*). PI(4,5)P$_2$ can also recruit mammalian SEPT4 onto the plasma membrane (*Zhang et al., 1999*), at least in vitro. We thus tested whether PI(4,5)P$_2$ also affects myelin septin assembly in vivo. To this aim, we re-investigated $Pten^{flox/flox};Cnp^{Cre/WT}$ mice (*Goebbels et al., 2010*), in which reduced PI(4,5)P$_2$–levels in myelin (*Figure 5A–B*) cause myelin outfoldings (*Figure 5C*) via a yet unknown mechanism. Indeed, the abundance of septins (SEPT2, SEPT4, SEPT7, SEPT8) and ANLN was strongly reduced in myelin purified from the brains of $Pten^{flox/flox};Cnp^{Cre/WT}$ mice (*Figure 5D*). This implies that the PI(4,5)P$_2$–dependent membrane-recruitment of ANLN and septins is principally conserved between the yeast cleavage furrow and murine myelin.

## Mice lacking a myelin septin/anillin filament

To examine whether a causal relationship exists between myelin septins and myelin outfoldings, we generated mouse mutants by gene targeting (*Figure 6—figure supplement 1A*). Promotor activity of the *Sept8*-gene in brains was monitored by lacZ histochemistry in $Sept8^{lacZ/WT}$ mice (*Figure 6—figure supplement 1C*). LacZ-staining was observed in grey and white matter areas (*Figure 6—figure supplement 1C*) and presumably not restricted to oligodendrocytes. This prompted us to generate mice lacking SEPT8 selectively from myelinating cells ($Sept8^{flox/flox};Cnp^{Cre/WT}$, also termed conditional mutants, COND) in addition to mice lacking SEPT8 from all cells ($Sept8^{null/null}$, also termed constitutive knockout, KO) (*Figure 6—figure supplement 1C*). Conditional and constitutive *Sept8*-mutants were born at expected frequencies, and their major myelinated fibre tracts formed normally as judged by light microscopic examination of histochemically stained myelin (*Figure 6A*). Electron microscopy did not reveal abnormalities of developmental myelination (*Figure 6B*), the

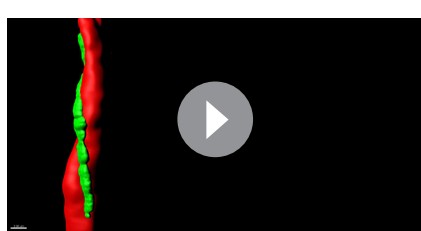

**Video 1.** 3-dimensional reconstruction of a myelin septin filament (SEPT8 immunolabelling, green) alongside an axon (neurofilament immunolabelling, red) Related to *Figure 2*.

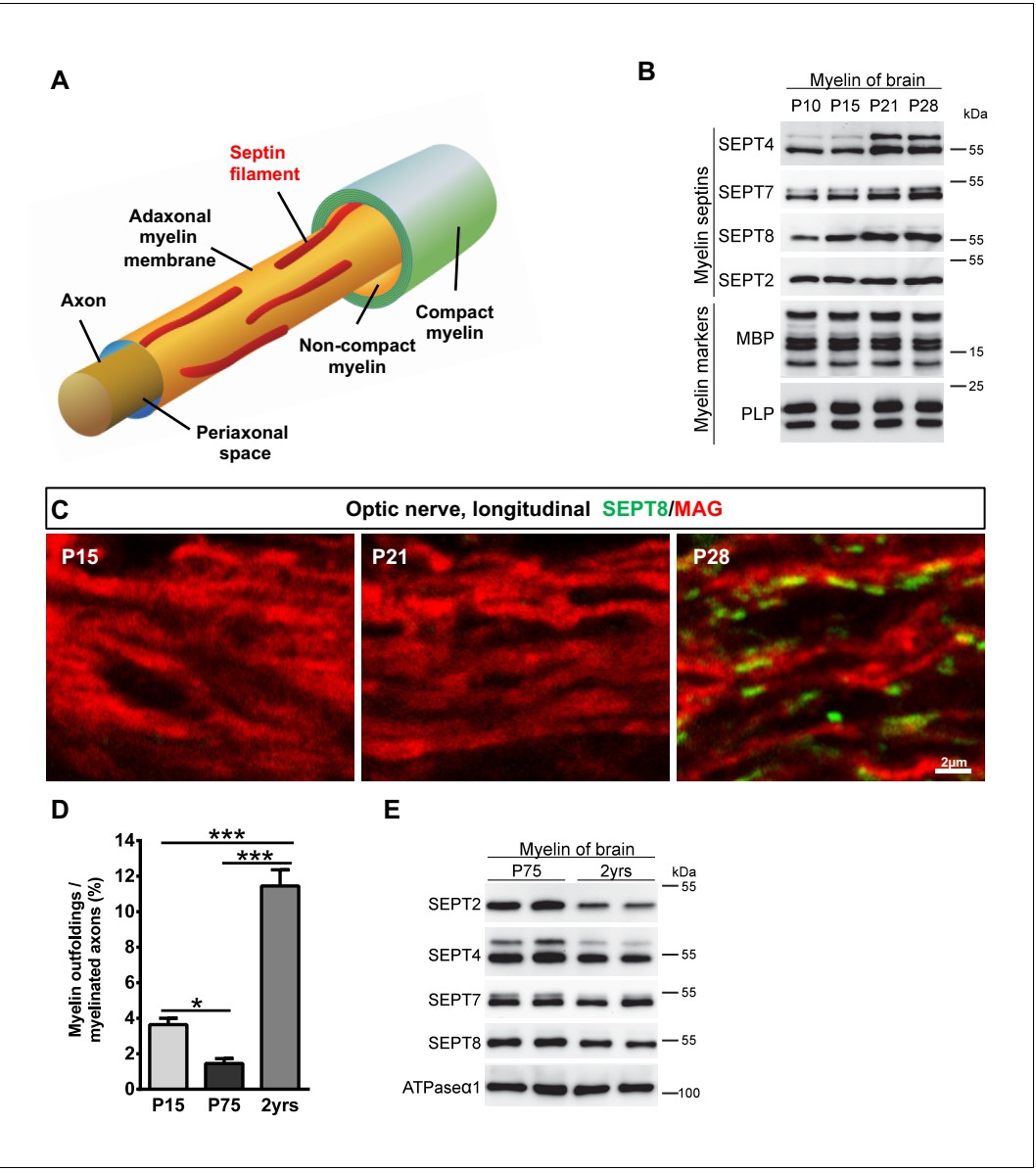

**Figure 3.** Septins in myelin development and aging. (**A**) Scheme showing the localization of septin filaments (red) in healthy adaxonal myelin. Note that all experimental data support a model that myelin septins (SEPT2, SEPT4, SEPT7, SEPT8) assemble as longitudinal filaments in the non-compact adaxonal compartment of myelin, in which they underlie the innermost membrane of compact myelin. (**B**) Immunoblotting indicates that the abundance of several septins (SEPT4, SEPT7, SEPT8) increases with age in myelin purified from wild-type brains at P10, P15, P21, and P28, reflecting the maturation of myelin. Note that the abundance of the classical myelin markers MBP and PLP is unaltered. (**C**) Immunolabelling of WT optic nerves detects SEPT8 (green) at P28 but not at P15 or P21. Note that the myelin marker MAG (red) is detectable at all time points, reflecting that the optic nerve is largely myelinated by P14. Images are representative of three experiments. (**D**) Quantitative evaluation of electron micrographs of WT optic nerves indicates that the frequency of axon/myelin-units with myelin outfoldings declines between P15 and P75 (i.e. with myelin maturation) and is strongly increased at 2 years of age (i.e with normal aging). Mean ± SEM. n=4 animals were analyzed. One-way ANOVA with Tukey's multiple comparison test; P15 vs. P75, *p=0.044; P15 vs. 2 yrs, ***p<0.001, P75 vs. 2 yrs, ***p<0.001. (**E**) Immunoblotting indicates that the abundance of myelin septins is decreased in myelin purified from mice at the age of two years compared to P75, reflecting normal aging. Blot is representative of 3 animals per age. ATPase-alpha1 (ATP1A1) served as a control.

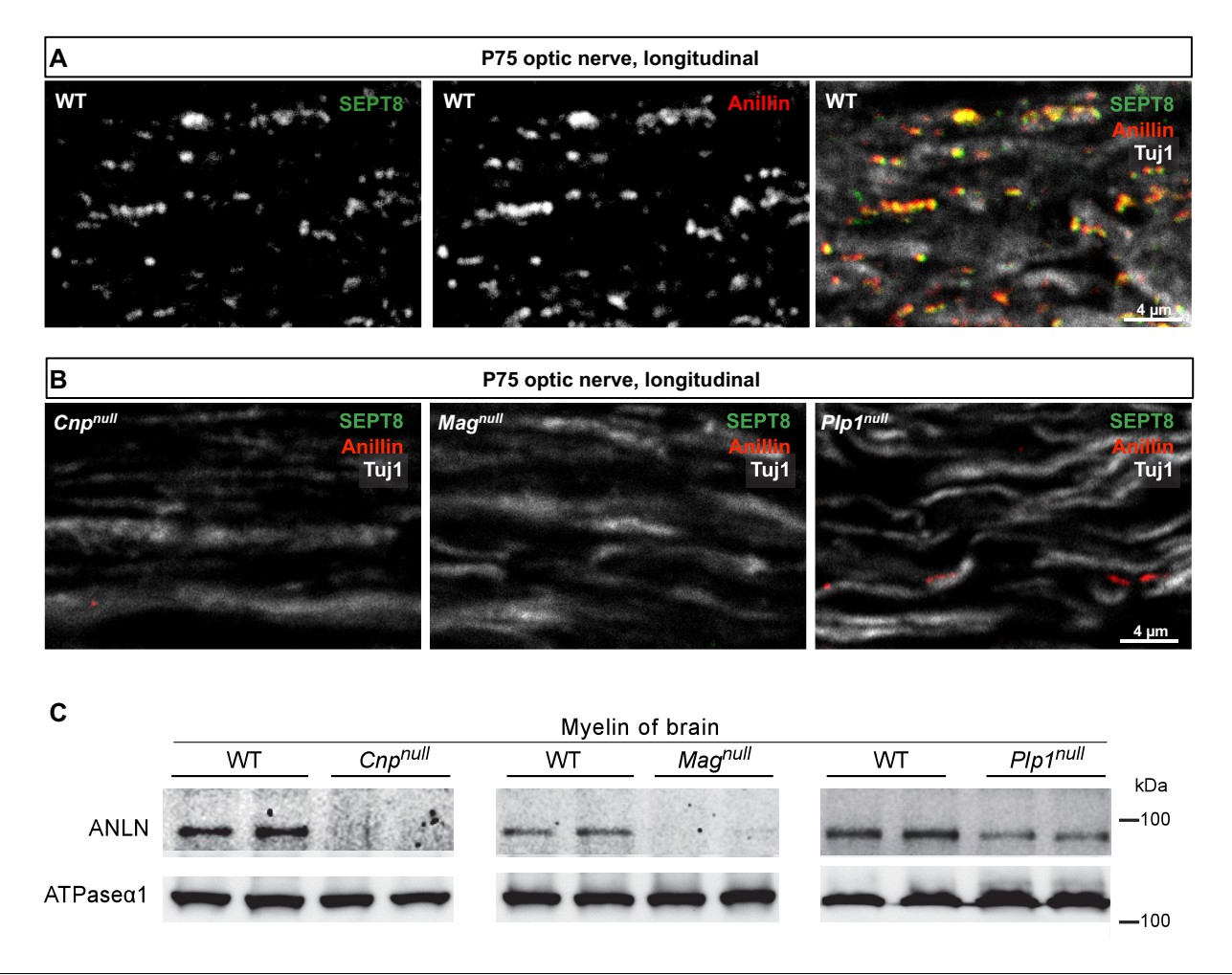

**Figure 4.** Anillin co-distributes with myelin septins. (**A**) Confocal microscopy reveals proximity-labelling of SEPT8 (green) and anillin (red). The merge additionally shows the axonal marker Tuj1 (white). Longitudinally sectioned optic nerves of P75 WT mice are shown. Example is representative of three independent experiments. (**B**) Immunolabelling indicates diminishment of anillin (red) in myelinated fibre tracts of *Cnp*^*null*^, *Mag*^*null*^, and *Plp1*^*null*^-mice similar to SEPT8 (green). The axonal marker TUJ1 (white) was co-labelled as a control. Longitudinally sectioned optic nerves of P75 mice are shown. Examples are representative of three independent experiments. (**C**) Immunoblotting of myelin purified from brains at P75 shows diminished abundance in *Cnp*^*null*^, *Mag*^*null*^, and *Plp1*^*null*^-mice compared to WT controls. ATPase-alpha1 (ATP1A1) served as a control. Blot is representative of n=3 animals.

percentage of myelinated axons (*Figure 6C*), and myelin thickness (*Figure 6D*).

Next, we examined if SEPT8-deficiency affects the protein composition of myelin. By label-free quantitative mass spectrometry, septins and ANLN were almost undetectable in myelin purified from *Sept8*^*null/null*^ mice whereas the abundance of all conventional myelin markers was unaltered compared to littermate controls (*Figure 6E*; *Figure 1—source data 1*). Loss of myelin septins in *Sept8*-mutant mice was validated by immunoblotting (*Figure 6F*; *Figure 6—figure supplement 1E*) and immunohistochemistry (*Figure 6—figure supplement 1F,G*). By immunoblotting, abundance and phosphorylation of Akt and Erk were similar in *Sept8*^*null/null*^ and control corpus callosi (data not shown). By qRT-PCR, cDNA-fragments for *Sept2, Sept4*, and *Sept7*, were amplified with about equal efficiency from mutant and control corpus callosi (*Figure 6—figure supplement 1D*), implying that the loss of myelin septins (secondary to SEPT8-deficiency) is a posttranscriptional event. Together, these results strongly suggest that SEPT2, SEPT4, SEPT7, and ANLN are degraded if not stabilized by their incorporation into a myelin septin filament, which requires SEPT8 in its core oligomer.

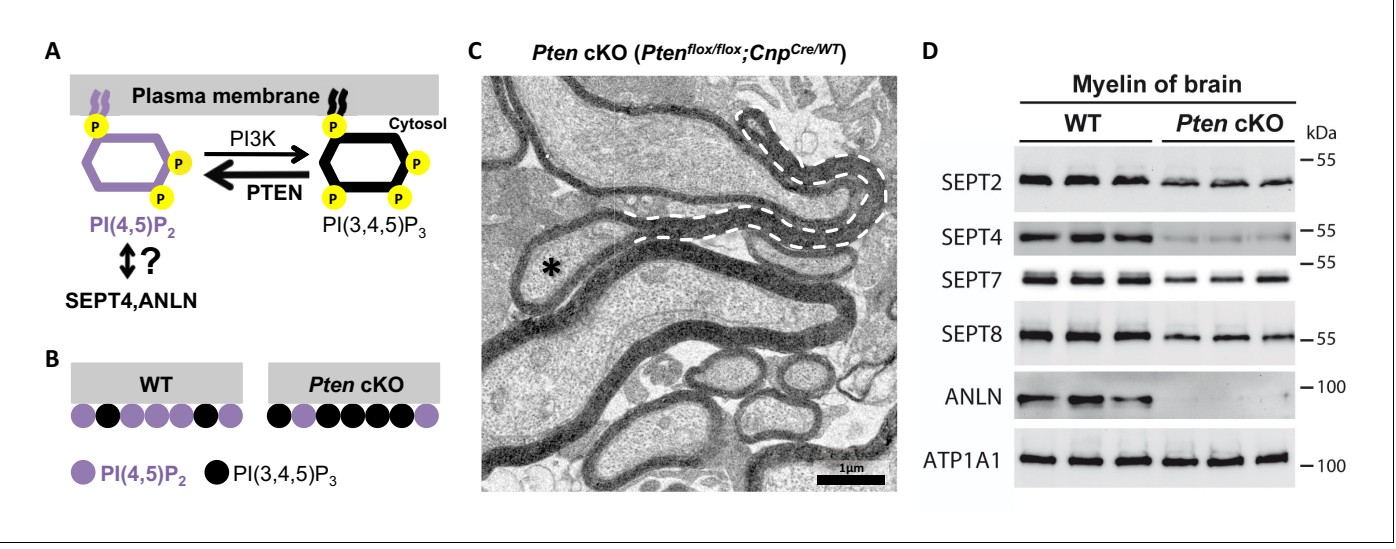

**Figure 5.** Membrane phosphoinositides mediate septin/anillin assembly in myelin. (**A**) Scheme of the reaction catalyzed by the phosphatase PTEN, which converts $PI(3,4,5)P_3$ into $PI(4,5)P_2$. The reverse reaction is mediated by phosphatidylinositol-3-kinase (PI3K). Previous reports indicated that SEPT4 and ANLN can be recruited onto membranes via $PI(4,5)P_2$, at least in vitro and in yeast, respectively; however an association has not yet been demonstrated in mice in vivo. (**B**) $Pten^{flox/flox};Cnp^{Cre}$ mice (Pten cKO) provide an established model in which the Cre-mediated deletion of Pten in myelinating cells causes a reduced abundance of $PI(4,5)P_2$ (**Goebbels et al., 2010**). (**C**) $Pten^{flox/flox};Cnp^{Cre}$ mice display myelin outfoldings (**Goebbels et al., 2010**). The electron micrograph of a P75 optic nerve cross-section showing several myelinated axons. A myelin outfolding and the associated axon are labelled with stippled line and asterisk, respectively. Image is representative of 4 animals per genotype. (**D**) Immunoblotting of myelin purified from mouse brains at P75 indicates that the abundance of myelin septins and anillin is reduced when oligodendroglial PTEN is lacking in $Pten^{flox/flox};Cnp^{Cre}$ mice (Pten cKO). The blot is representative of 3 animals per genotype. ATP1A1 served as a control.

When analyzing mature *Sept8*-mutants by electron microscopy, myelin outfoldings were obvious (*Figure 7A*). However, myelin outfoldings were a significant feature only in adult *Sept8*-mutants, but not at the developmental age of P15 (*Figure 7B*). We did not observe hypomyelination (*Figure 6C, D*), axonal spheroids (not shown), degenerated axons (*Figure 7C*), inner tongue swellings (*Figure 7E*), astrogliosis (*Figure 7—figure supplement 1A*), or microgliosis (*Figure 7—figure supplement 1B*). Thus, the deficiency of myelin septin filaments very specifically caused myelin outfoldings but not complex neuropathology, in marked difference to the lack of CNP, MAG, PLP, or oligodendroglial PTEN. It is noteworthy that myelin outfoldings per se do not cause neuroinflammation.

To elucidate the dynamics of the degradation of the septin/anillin scaffold in mature myelin, we have induced recombination of the *Sept8* gene by injecting Tamoxifen into adult $Sept8^{flox/flox};Plp^{Cre-ERT2}$ mice (*Figure 7—figure supplement 2A*), thereby preventing the assembly of new myelin septin core oligomers in myelinating cells. $Sept8^{flox/flox}$ mice subjected to Tamoxifen injections served as controls. By immunoblot, the abundance of myelin septins (SEPT2, SEPT4, SEPT7, SEPT8) was about halved in myelin purified from the brains of $Sept8^{flox/flox};Plp^{Cre-ERT2}$ mice 4 weeks post Tamoxifen injection (pti) and greatly reduced 8 weeks pti (*Figure 7—figure supplement 2B*), reflecting the degradation of myelin septin filaments. Using the same model we also tested whether depleting the septin/anillin-scaffold causes myelin outfoldings when induced after developmental myelination has been completed. Indeed, myelin outfoldings were evident 18 weeks pti according to electron microscopic analysis (*Figure 7—figure supplement 2C,D*). Thus, continuous replenishment of the myelin septin scaffold is required to prevent the formation of myelin outfoldings.

## Analysis of nerve conduction velocity

To test whether the septin/anillin-scaffold of myelin is relevant for CNS function in vivo, we measured nerve conduction in the spinal cord of $Sept8^{null/null}$ and $Sept8^{flox/flox};Cnp^{Cre/WT}$ mice at various ages. Indeed, conduction velocity was reduced by about 20% in *Sept8*-mutants (*Figure 7E*). Considering

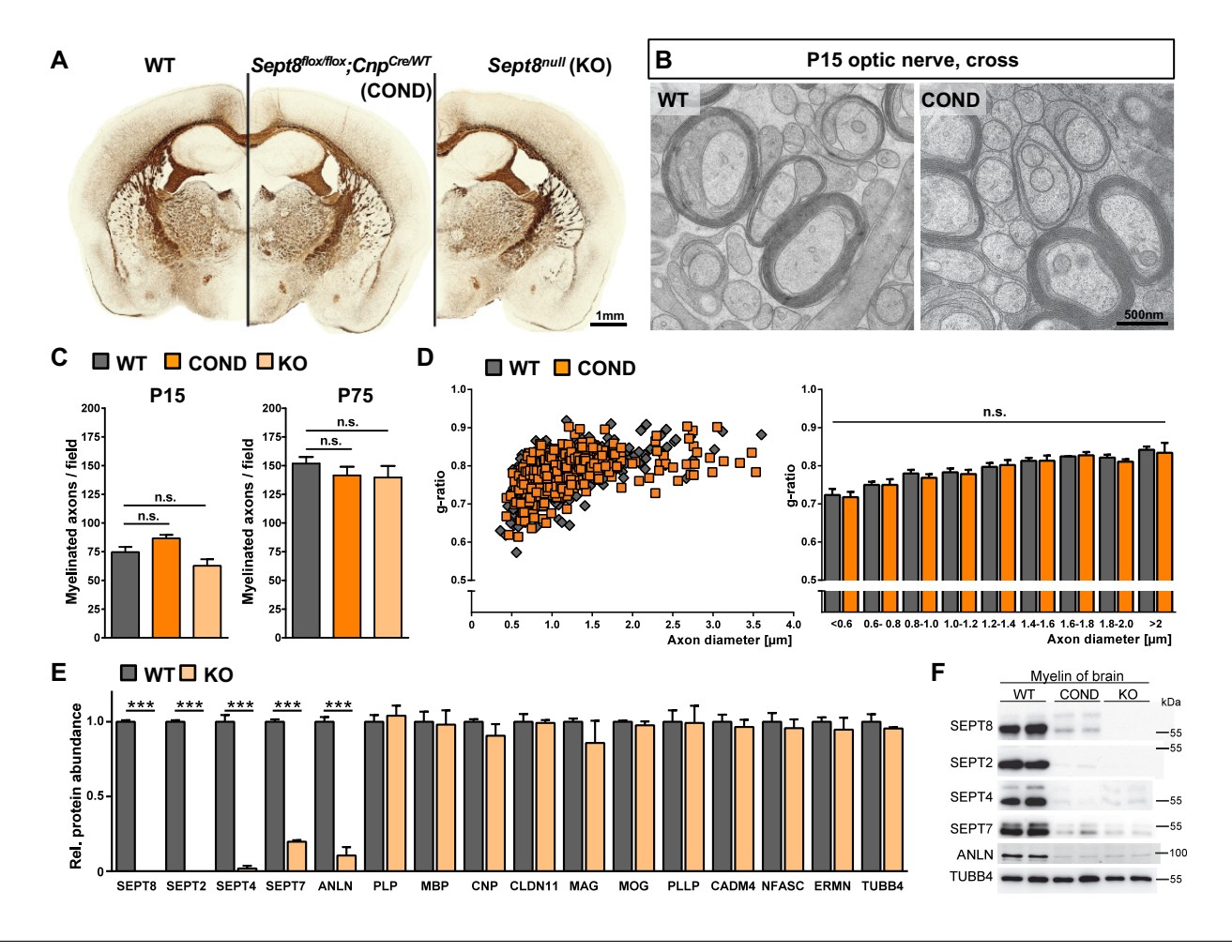

**Figure 6.** Myelination in mice lacking myelin-associated septins. (A) Silver impregnation visualizes myelinated fibre tracts in mice lacking SEPT8 from myelinating cells (*Sept8^flox/flox*;*Cnp^Cre/WT*-mice; COND) or from all cells (*Sept8^null/null*-mice; KO). Images are representative of three animals per genotype. See *Figure 6—figure supplement 1* for generation and validation of *Sept8* mutant mice. (B) Electron micrographs of cross-sectioned WT and *Sept8^flox/flox*;*Cnp^Cre/WT*-mice (COND) optic nerves fixed by high-pressure freezing indicate indistinguishable myelin ultrastructure at P15. Images are representative of three animals per genotype. (C) Quantitative evaluation of electron micrographs of optic nerves at P15 and P75 reveals a normal frequency of myelinated axons in *Sept8*-mutant mice (COND, KO). Mean +/ SEM. n=5 animals per condition; not significant (n.s.) according to one-way ANOVA with Tukey's post test; see Materials and methods section for p-values. (D) g-ratio analysis of electron micrographs of optic nerves at P75 indicates normal myelin sheath thickness in *Sept8^flox/flox*;*Cnp^Cre/WT*-mice (COND). Mean +/ SEM. Not significant (n.s.) according to two-way ANOVA (p=0.7823). (E) Differential myelin proteome analysis reveals that septins (SEPT2, SEPT4, SEPT7) and anillin (ANLN) are diminished in myelin purified from *Sept8^null/null*-mice (KO) at P75, whereas classical myelin proteins are not affected. Mean +/ SEM. n=3 animals per genotype; two-tailed unpaired t-test ***p<0.001. See *Figure 1—source data 1* for the entire dataset. (F) Immunoblotting validates the lack of myelin septins and anillin (ANLN) in myelin purified from the brains of *Sept8*-mutant mice (COND, KO). Tubulin-beta4 (TUBB4) was detected as a control. The blot is representative of three experiments.

The following figure supplement is available for figure 6:

**Figure supplement 1.** Generation of mice lacking expression of SEPT8.

that the density of nodes of Ranvier is unaltered in *Sept8^null/null* mice (**Figure 7—figure supplement 1C,D**), this finding suggests that myelin outfoldings affect saltatory conduction. While it is plausible that myelin outfoldings impede current flow along myelinated fibre tracts, we cannot formally rule out that unidentified secondary effects also contribute to decelerated conduction.

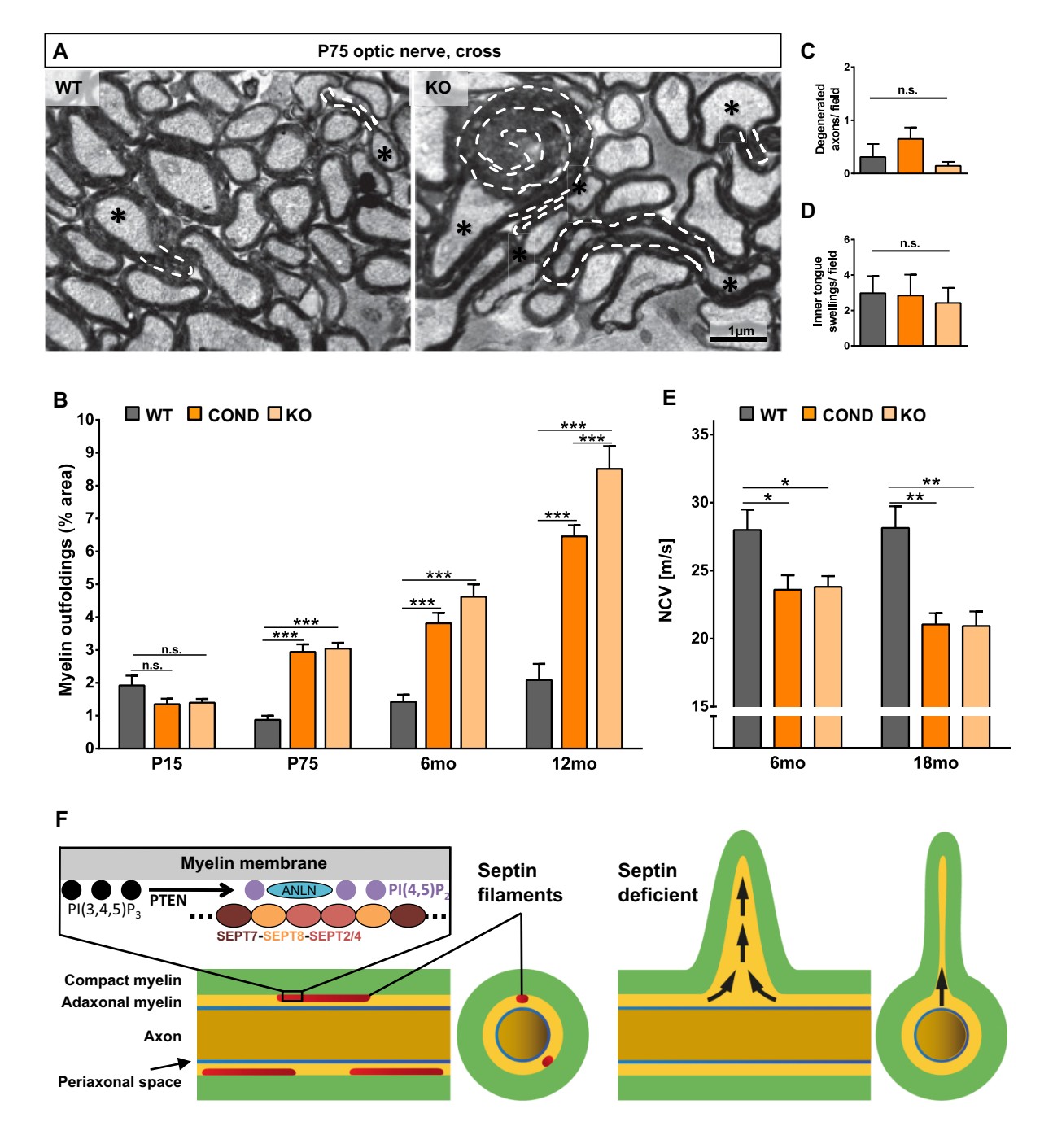

**Figure 7.** Lack of myelin septins causes myelin outfoldings and decelerated nerve conduction. (**A**) Electron micrographs of optic nerves exemplify myelin outfoldings at P75. Stippled lines highlight myelin outfoldings; associated axons are marked with asterisks. (**B**) Quantitative evaluation of optic nerve electron micrographs reveals progressive myelin outfoldings in adult *Sept8*-mutant mice (COND, KO). Mean +/ SEM. n=3–6 animals per condition; significant according to two-way ANOVA for the effects of genotype (p<0.0001) and age (p<0.0001); see Materials and methods section for p-values of Tukey's post-test. (**C–D**) Quantitative evaluation of electron micrographs of optic nerves at age P75 indicates the absence in *Sept8*-mutant mice (COND, KO) of other myelin-related pathology such as degenerated axons (**C**) or inner tongue swellings (**D**). Mean +/ SEM. n=3 animals; not significant (n.s.) according to one-way ANOVA; degenerated axons p=0.25 (**C**), inner tongue swellings p=0.92 (**D**). (**E**) Electrophysiological measurement reveals decelerated nerve conduction in spinal cords of *Sept8*-mutant mice (COND, KO) compared to controls at 6 and 18 months of age. Mean +/ SEM. n=4–9 animals per genotype and age; two-way ANOVA shows the significance for genotype-dependency of the effect (p<0.0001); for p-values of Tukey's post-test see Materials and methods section. (**F**) Hypothetical model of pathological outfoldings of compact myelin (green) upon loss of myelin

*Figure 7 continued on next page*

*Figure 7 continued*

septins. Shown are longitudinal and cross-sectional views. Arrows indicate hydrostatic outward pressure and cytoplasm flow in adaxonal myelin (orange) unless stabilized by septin filaments (red). Inset, the septin/anillin scaffold is assembled from core-hexamers in dependence of PI(4,5)$P_2$-levels.

The following figure supplements are available for figure 7:

**Figure supplement 1.** SEPT8 deficiency does not induce astrogliosis, microgliosis or altered structure or density of the nodes of Ranvier.

**Figure supplement 2.** Dynamics of septin filament degradation in myelin and emergence of myelin outfoldings in adult mice.

## Discussion

Myelin outfoldings are frequently observed during developmental myelination, and these membranes have been interpreted as a reservoir for the elongation of axon/myelin-units during body growth (*Cullen and Webster, 1979*; *Rosenbluth, 1966*; *Snaidero et al., 2014*). Our data reveal that the assembly of septin filaments is a late stage of myelin maturation. By forming a scaffold underlying the innermost membrane layer of compact myelin, septins prevent outfoldings of the entire stack of compact myelin layers in the adult CNS. Septin filaments assemble by associating with membranes (*Bridges et al., 2014*) involving the interactions of particular septin monomers and the adaptor protein anillin with PI(4,5)$P_2$ (*Bertin et al., 2010*; *Liu et al., 2012*; *Zhang et al., 1999*), a recruitment mechanism conserved between the yeast cleavage furrow and mammalian myelin. As submembraneous filaments, septins can provide structure and rigidity to the membranes they are associated with (*Gilden and Krummel, 2010*; *Spiliotis and Gladfelter, 2012*; *Tooley et al., 2009*). The adaxonal myelin compartment is regularly spaced, and the cytoplasmic surfaces of the flanking membranes are in close proximity but not compacted. Adaxonal myelin differs from compact myelin by lacking MBP and instead containing oligodendroglial cytoplasm. In our model (*Figure 7F*), the septin/anillin scaffold stabilizes compact myelin by associating with its innermost membrane surface. This architecture counteracts hydrostatic pressure and lateral membrane flow (*Gilden and Krummel, 2010*), which in the absence of adhesive forces would push compact myelin outward, e.g. as a consequence of the net growth of individual myelin membranes or the slow but lifelong intercalation of new myelin segments between existing sheaths (*Young et al., 2013*).

High PI(3,4,5)$P_3$-levels trigger the active net growth of myelin sheaths and thus myelin thickness via the Akt/mTOR pathway (*Chan et al., 2014*; *Goebbels et al., 2010*; *Macklin, 2010*) whereas high PI(4,5)$P_2$-levels are required to recruit the septin/anillin-scaffold of myelin. Loss of septins in the myelin sheath is likely owing to dysregulated filament assembly or stability, which in addition to membrane-phosphoinositides and ANLN may also involve other regulatory factors (*Joberty et al., 2001*). Several mechanisms are thus feasible how mutations affecting structural myelin proteins may cause the post-transcriptional decrease of septins. For example, PLP is required for a normal myelin lipid composition (*Werner et al., 2013*) and CNP prevents premature closure of the cytoplasmic channels through myelin (*Snaidero et al., 2014*), which probably provide transport routes for molecules and vesicles into the adaxonal myelin layer (*Nave and Werner, 2014*). It is plausible that both phenomena may directly or indirectly affect the septin/anillin-scaffold that prevents the formation of myelin outfoldings.

The axon/myelin-unit comprises various stabilizing molecules. Recently, a cytoskeletal actin/spectrin-lattice has been discovered that underlies the axonal membrane along its entire length (*Xu et al., 2013*). Moreover, neuronal and glial Ig-CAMs interact along the axon-myelin interface (*Kinter et al., 2012*). We note that at the sites of pathological myelin outfoldings, the axo-myelinic membrane apposition is not disrupted. However, the lack of myelin septins causes the entire stack of compacted myelin to focally detach from the adaxonal oligodendroglial membrane and to 'fold out'. This reflects strong adhesive forces between adjacent layers of compact myelin mediated by myelin proteins such as MBP. The non-compact compartments of myelin are considered as cytoplasmic routes enabling the axon-supportive function of oligodendrocytes (*Nave and Werner, 2014*). We conclude that the septin/anillin-scaffold reported here stabilizes mature CNS myelin while allowing diffusion and transport processes in the non-compact adaxonal myelin layer to take place.

## Materials and methods

### Mouse models

$Plp1^{null}$, $Cnp^{null}$, $Mag^{null}$, and $Pten^{flox/flox};Cnp^{Cre/WT}$ mice were described previously (*Goebbels et al., 2010*; *Klugmann et al., 1997*; *Lappe-Siefke et al., 2003*; *Montag et al., 1994*). To homogenize the genetic background, the lines were backcrossed to the C57BL/6N strain for >10 generations before breeding the experimental animals for the present study. Genotyping of the *Plp1* allele was by genomic PCR with primers 1864 (5'-TTGGCGGCGA ATGGGCTGAC), 2729 (5'-GGAGAGGAGG AGGGA-AACGAG), and 2731 (5'-TCTGTTTTGC GGCTGACTTTG). PCR genotyping of the *Cnp* allele was with primers 2016 (5´-GCCTTCAAAC TGTCCATCTC), 7315 (5´-CCCAGCCCTT TTATTACCAC), 4193 (5´-CCTGGAAAAT GCTTCTGTCCG), and 4192 (5´-CAGGGTGTTA TAAGCAATCCC). PCR genotyping of the *Mag* allele was with primers 1864 (5'-TTGGCGGCGA ATGGGCTGAC), 7650 (5'-ACGGC-AGGGA ATGGAGACAC), and 7649 (5'-ACCCTGCCGC TGTTTTGGAT). PCR genotyping of the *Pten* allele was with primers 5495 (5´-ACTCAAGGCA GGGATGAGC) and 20515 (5´-CAGAGTTAAG TTTTTGAAGGCAAG).

Embryonic stem cells (ES) harbouring an engineered allele of the *Sept8* gene were acquired from the European Conditional Mouse Mutagenesis Program (Eucomm). ES were microinjected into blastocysts derived from FVB mice, and embryos were transferred to pseudo-pregnant foster mothers, yielding 6 chimeric males. For ES clone EPD0060_3_G04, germline transmission was achieved upon breeding with C57BL/6N-females, yielding mice harbouring the $Sept8^{lacZ-neo}$ allele. The lacZ/neo cassette was excised in vivo upon interbreeding with mice expressing FLIP recombinase (129S4/SvJae-Sor-Gt(ROSA)26Sortm1(FLP1)Dym/J; backcrossed into C57BL/6N), yielding mice carrying a $Sept8^{flox}$ allele. To inactivate expression of SEPT8 whole-body-wide, exon 2 of the *Sept8* gene was excised in vivo upon interbreeding with mice expressing Cre recombinase under control of the adenoviral *EIIA* promoter (*Holzenberger et al., 2000*) (backcrossed into C57BL/6N), yielding mice carrying a $Sept8^{null}$ allele. When heterozygous $Sept8^{null}$ mice were interbred to obtain homozygous mutants, those were born at Mendelian frequency. For simplicity, $Sept8^{null/null}$ mice are also termed knockout ('KO'). To inactivate expression of SEPT8 in myelinating cells, exon 2 was excised in vivo upon appropriate interbreedings of $Sept8^{flox}$ mice with mice expressing Cre recombinase under control of the *Cnp* promoter (*Lappe-Siefke et al., 2003*). For simplicity, $Sept8^{flox/flox};Cnp^{Cre/WT}$ mice are also termed conditional mutant ('COND'). Corresponding control mice (genotypes $Sept8^{WT/WT};Cnp^{WT/WT}$, $Sept8^{flox/WT};Cnp^{WT/WT}$, and $Sept8^{flox/flox};Cnp^{WT/WT}$) are labelled 'WT 'throughout. For Tamoxifen-induced inactivation of SEPT8 expression in myelinating cells, $Sept8^{flox}$ mice were interbred with mice expressing Cre-ERT2 in myelinating cells under control of the *Plp*-promoter (*Leone et al., 2003*). $Sept8^{flox/flox};Plp^{Cre-ERT2}$ (inducible conditional mutant; *Sept8* icKO) and control $Sept8^{flox/flox}$ mice were injected with 1 mg Tamoxifen per day beginning at 8 weeks of age for five consecutive days, then after a pause of two days again for five consecutive days.

Routine genotyping of the *Sept8* allele was by PCR with sense primer P1 (5'-GAAGCAGCCA TA-GAGGAGATCC; binding 5 'of the first loxP-site) in combination with antisense primers P2 (5'-GGTG-GCTTTG AACTTGCTATCC; binding the segment flanked by loxP-sites), and P3 (5'-CAGGCAGATG TATATGCAGCAG; binding to the lacZ cassette) or P4 (5'-CAGGCAGATG TATATGCAGCAG; binding 3 'of the third loxP site). The PCR shown in *Figure 6—figure supplement 1B* was performed with P1, P2, and P3. All experimental animals were progeny of heterozygous parents to facilitate that mutant animals were analyzed together with littermate controls as far as possible. Mice were kept in the mouse facility of the Max-Planck-Institute of Experimental Medicine with a 12 hr light/dark cycle and 2–6 animals per cage. All experiments were approved by the local Animal Care and Use Committee in agreement with the German Animal Protection Law.

### Quantifications and statistical analysis

All quantifications were performed blinded to the genotypes. All graphs display mean values and standard error of the mean (SEM) as error bars. All statistical tests were performed using GraphPad Prism 6.0. Tests were chosen depending on experimental groups and as suggested by the software. To test for variance, F-test was performed using GraphPad Prism 6.0. GraphPad online test was used to detect outliers (http://graphpad.com/quickcalcs/Grubbs1.cfm); however no outliers were identified. The levels of significance were set at $p < 0.05$ (*), $p < 0.01$ (**), and $p < 0.001$ (***).

For quantifications displayed in *Figure 6C*, p-values are as follows: P15: WT vs. COND p=0.19, WT vs. KO p=0.19; P75: WT vs. COND p=0.64, WT vs. KO p=0.55. For quantifications displayed in *Figure 7B*, p-values for genotype-dependent comparisons are as follows: P15: WT vs. COND p=0.29, WT vs. KO p=0.35, COND vs. KO p=0.99; P75: WT vs. COND p<0.0001, WT vs. KO p<0.0001, COND vs. KO p=0.96; 6 mo: WT vs. COND p<0.0001, WT vs. KO p<0.0001, COND vs. KO p=0.14; 12 mo: WT vs. COND p<0.0001, WT vs. KO p<0.0001, COND vs. KO p<0.001. p-values for age-dependent comparisons in *Figure 7B* are as follows: WT: P75 vs. P15 p=0.04, 6mo vs. P15 p=0.5074, 12mo vs. P15 p=0.98, 6mo vs. P75 p=0.43, 12mo vs. P75 p=0.04, 12mo vs. 6mo p=0.39; COND: P75 vs. P15 p<0.001, 6mo vs. P15 p<0.0001, 12mo vs. P15 p<0.0001, 6mo vs. P75 p=0.12, 12mo vs. P75 p<0.0001, 12mo vs. 6mo p<0.0001; KO: P75 vs. P15 p<0.001, 6mo vs. P15 p<0.0001, 12mo vs. P15 p<0.0001, 6mo vs. P75 p=0.003, 12mo vs. P75 p<0.0001, 12mo vs. 6mo p<0.0001. For quantifications displayed in *Figure 7E*, two-way ANOVA shows the significance for genotype-dependency of the effect (p<0.0001), but not for age-dependency (p=0.113). p-values of Tukey's multiple comparison tests are as follows: 6mo: WT vs. COND p=0.03, WT vs. KO p=0.04, COND vs. KO p=0.99; 18mo: WT vs. COND p=0.006, WT vs. KO p=0.002, COND vs. KO p=0.998. For other p-values, see the respective figure legends.

## Electron microscopy (EM)

Immungold labelling of cryosections was performed as described (*Werner et al., 2007*). Optic nerves dissected from male WT (C57Bl/6N) mice at P75 were used. Antibodies were specific for SEPT8 (ProteinTech Group 11769-1-AP, 1:50) (yielding similar results as a previously published antibody (*Ihara et al., 2003*) kindly provided by M. Kinoshita, 1:50), SEPT7 (IBL18991, 1:50), or MAG (Chemicon MAB1567, 1:50). To quantitatively assess the exact protein localization within the adaxonal non-compact myelin layer, all gold particles located to this compartment on the micrographs (a minimum of 200 gold particles per animal) were assigned to one of two categories as schematically depicted in *Figure 2H'*: associated with the innermost membrane layer of compact myelin (type 1) or associated with the adaxonal myelin membrane (type 2).

Preparation of samples for transmission electron microscopy by high pressure freezing and freeze substitution was performed essentially as described (*Möbius et al., 2010*). Briefly, optic nerves were dissected and placed into aluminum specimen carriers with an indentation of 0.2 mm. The remaining space was covered with 20% PVP (Sigma) in PBS, and the sample was cryofixed using a HPM100 high-pressure freezer (Leica). Freeze substitution was carried out in a Leica AFS (Leica, Vienna, Austria) as follows: samples were initially kept in tannic acid (0.1% in acetone) at −90°C for 100 hr, washed with acetone (4 × 30 min, −90°C) and then transferred into $OsO_4$ (EMS; 2%) and uranyl acetate (SPI Chem, 0.1%) in acetone at −90°C. The temperature was raised from −90 to −20°C in increments of 5°C/hr, then kept unaltered at −20°C for 16 hr, and then raised to +4°C in increments of 10°C/hr. The samples were then washed with acetone (3 × 30 min at 4°C), allowed to adjust to room temperature, and finally transferred into Epon (Serva) (25%, 50%, and 75% Epon in acetone for 1–2 hr each, 90% Epon in acetone for 18 hr, 100% Epon for 4 hr). The samples were placed in an embedding mold and polymerized (60°C, 24 hr). Preparation of samples for transmission electron microscopy by conventional aldehyde fixation was performed as described (*Möbius et al., 2010*). Briefly, mice were perfused with 2,5% glutaraldehyde and 4% formaldehyde in phosphate buffer containing 0.5% NaCl. Spinal cord samples, optic nerves, and brain samples were postfixed in 1% $OsO_4$ in 0.1 M phosphate buffer and embedded in epoxy resin (Serva). Ultrathin sections (50 nm) were cut using a Leica Ultracut S ultramicrotome (Leica) and contrasted with an aqueous solution of 4% uranyl acetate (SPI Chem) followed by lead citrate. The samples were examined in a LEO 912AB Omega transmission electron microscope (Zeiss). Pictures were taken with an on-axis 2048 × 2048-CCD-camera (TRS).

For assessment of pathology, mice were analyzed at postnatal day 75 (P75) unless indicated otherwise; 3–6 male mice were used per genotype and age. 10–15 randomly selected, non-overlapping images were taken per optic nerve at 7000 × magnification (1 field=220 µm). Electron micrographs were analyzed using ImageJ (Fiji). To quantify axonal pathology and the proportion of nonmyelinated axons, all axons on the micrographs (a minimum of 700 axons per animal) were assigned to one of five categories: healthy-appearing myelinated axons, healthy-appearing nonmyelinated axons, axons with myelin comprising a swollen adaxonal compartment (inner tongue), axonal spheroids, and degenerated axons. Axons were counted as myelinated if ensheathed by at least one complete

layer of compacted myelin. Axonal changes were identified by mild signs of pathology including invaginations of adaxonal myelin membrane into the axon. Degenerated axons were identified by tubovesicular structures and amorphous cytoplasm. The area occupied by myelin outfoldings was quantified by a point counting method (*Edgar et al., 2009*). Briefly, a regular grid of 0.25 μm$^2$ was placed on the images. The number of intercepts coinciding with myelin outfoldings was related to the evaluated area. When quantifying myelinated axons, axonal pathology, axonal diameters, and myelin outfoldings, to compare *Cnp^null*, *Mag^null*, and *Plp1^null* mice with WT mice significance was determined using GraphPad Prism 6.0. To compare SEPT8 mutant mice with WT littermates, significance was determined using GraphPad Prism 6.0. The g-ratio was calculated as the ratio between the axonal Feret diameter and the Feret diameter of the corresponding myelin sheath. To determine g-ratios, a regular grid was placed on the images for randomization. All axons crossed by the grating were assessed, yielding a minimum of 100 myelinated axons per animal. The g-ratio was assessed using GraphPad Prism 6.0. All quantifications were performed blinded to the genotype.

## Immunohistochemistry

Immunohistochemistry to determine neuropathology on sections of paraffin-embedded brains and spinal cords was essentially as described (*de Monasterio-Schrader et al., 2013*). Antibodies were specific for MAC3 (1:400; Pharmingen) or glial fibrillary acidic protein (GFAP; 1:200; NovoCastra). Images were captured at 20x magnification using a bright-field light microscope (Zeiss AxioImager Z1) coupled to a Zeiss AxioCam MRc camera controlled by Zeiss ZEN 1.0 software and processed using ImageJ 1.46 and Adobe Photoshop. Mice were 12 months old; sections from 4–6 male mice per genotype including the mean of both fimbriae were quantified. To quantify white matter area immunopositive for MAC3 or GFAP, the hippocampal fimbria was selected and analyzed using an ImageJ plugin for semiautomated analysis. Data were normalized to wild-type levels. All quantifications were performed blinded to the genotype. Statistical analysis was performed using GraphPad Prism 6.0.

Silver impregnation of myelin on histological sections was as described (*Gallyas, 1979*). Images were captured at 10x magnification (Zeiss AxioImager Z1) and stitched using Zeiss Zen2011. *Figure 6A* shows sections from mice of the indicated genotypes at one year of age.

To visualize cells with *Sept8* gene activity, we performed lacZ immunohistochemistry on the brains of heterozygous *Sept8^lacZ* mice. Mice were perfused with 4% PFA. Vibratome sections of 100 μm thickness (Leica VT 1000S) were incubated at 37°C with X-gal solution (1.2 mg X-gal per ml, 5 mM potassium ferricyanide, 5 mM potassium ferrocyanide, and 2 mM MgCl$_2$ in PBS). After the sections had incubated for about 2 hr in the dark, they were rinsed with PBS, dried, and mounted using Aqua-Poly/Mount (Polysciences). Images of the whole brain were captured at 10x magnification (Zeiss AxioImager Z1) and stitched (Zeiss Zen2011); the cerebellum was imaged at 40x magnification.

To determine the expression and localization of myelin septins, immunohistochemistry was performed on cryosectioned optic nerves and spinal cords. Blocking solution contained 10% horse serum, 0.5% triton X-100, and 1% BSA in PBS. Antibodies were specific for SEPT7 (IBL18991; 1:1000), SEPT8 (ProteinTech Group 11769-1-AP; 1:500), ANLN (Acris AP16165PU-N; 1:200), TUJ1 (Covance; 1:1000), neurofilament (SMI31; Covance; 1:1500), myelin-associated glycoprotein (MAG 513; Chemicon; 1:50), voltage-gated sodium channel Na$_v$1,6 (alomonelabs; 1:500), or contactin-associated protein (CASPR; Neuromabs; 1:500). Secondary antibodies were donkey α-rabbit-Alexa488 (dianova), donkey α-goat-Cy3 (dianova), and donkey α-mouse Dyelight633 (Yo-Pro). Images were obtained by confocal microscopy (Leica SP5). The signal was collected sequentially with the objective HCX PL APO CS 63.0 × 1.30 GLYC 21°C UV. An argon laser with the excitation of 488 nm was used to excite the Alexa488 fluorophore, and the emission was set to 500–573 nm. The laser DPSS 561 was used to excite the Cy3 fluorophore, and the emission was set to 573–630 nm. The HeNe laser 633 was used to excite Dyelight633, and emission was detected between 645–738 nm. DAPI was excited with 405 nm and collected between 417–480 nm. The LAS AF lite and Fiji were used to export the images as tif-files. Imaris was used for 3D-reconstructions.

The number of SEPT8-puncta (in *Figure 2D–F*) was determined using Fiji. Axonal diameter was determined using a threshold for the neurofilament immunolabelling signal and the particle analyzer plugin. SEPT8-punctae per myelinated axon as identified by the MAG-immunopositive ring were counted in WT at P75 (n=4 animals, 1 section each). Per animal, 6 random confocal micrographs of

spinal cord white matter with a size of 2500 µm² per micrograph were quantified, yielding about 100 axons per animal. Statistical analysis was performed using GraphPad Prism 6.0. For the quantifications of nodal density, the frequency of occurrence of two CASPR-immunopositive paranodes was analyzed using Fiji. CASPR-immunopositivity was converted using a threshold and counted using ITNC plugin (n=3 animals per genotype, 1 section each, 8 random confocal micrographs of spinal cord white matter with a size of 2500 µm² per micrograph). Statistical analysis was performed using GraphPad Prism 6.0.

## Myelin purification

A light-weight membrane fraction enriched for myelin was purified from mouse brains by sucrose density centrifugation and osmotic shocks as described (*Jahn et al., 2013*). For immunoblot analyses of myelin during development and aging, male wild-type (C57Bl/6N) mice of the indicated ages were used. For proteome or immunoblot analyses of mutant mice and their respective wild-type littermates, male mice at the age of 75 days were used. Protein concentrations were determined using the DC protein assay (BioRad).

## Proteome analysis

Differential quantitative myelin proteome analyses were based on previously described label-free, gel-free procedures (*Distler et al., 2014*; *Jahn et al., 2013*; *Patzig et al., 2011*). Briefly, brains were dissected from three male $Cnp^{null}$, $Mag^{null}$, $Plp1^{null}$, or $Sept8^{null/null}$ mice at the age of 75 days and three respective control littermates, and myelin was biochemically purified (see above). Per sample, 25 µg of total myelin protein was precipitated using the ProteoExtract Protein Precipitation Kit (Calbiochem), solubilized in 50 mM $NH_4HCO_3$ with 0.1% Rapigest (Waters) (10 min, 85°C), reduced with 5 mM DTT (45 min, 56°C), and alkylated with 15 mM iodoacetamide (45 min, room temperature) in the dark. Solubilized proteins were digested with 0.5 µg of sequencing grade trypsin (Promega) for 16 hr (37°C). After digestion, Rapigest was hydrolyzed by adding 10 mM HCl, the resulting precipitate was removed by centrifugation (13,000 g, 15 min, 4°C), and the supernatant was transferred into an autosampler vial. Data for the analysis of myelin purified from $Cnp^{null}$, $Plp1^{null}$, and $Sept8^{nul/null}$ mice and the respective littermate controls were obtained using separation of tryptic peptides by nanoscale ultraperformance liquid chromatography (nanoUPLC) coupled to a quadrupole time of flight (QTOF) Premier mass spectrometer (Waters) with an alternating low and elevated (E) energy mode of acquisition ($MS^E$) (UPLC-$MS^E$). Data for the analysis of myelin purified from $Mag^{null}$ mice and littermate controls were obtained using a Synapt G2S mass spectrometer (Waters). For each biological replicate, 2–4 technical replicates were measured. The continuum LC-$MS^E$ data were processed and searched using the IDENTITYE-algorithm of ProteinLynx Global Server (PLGS) version 2.3 (Waters). Protein identifications were assigned by searching the Uni-ProtKB/Swiss-Prot Protein Knowledgebase Release 52.3 for mouse proteins (12,920 entries) supplemented with known possible contaminants (porcine trypsin, human keratins) using the precursor and fragmentation data afforded by LC-MS acquisition as described previously (*Patzig et al., 2011*). Mass tolerances for peptide and fragment ions were set at 15 and 30 ppm, respectively. Peptide identifications were restricted to tryptic peptides with no more than one missed cleavage. Carboxyamidomethylation of Cys was set as fixed modification, and oxidation of Met, acetylation of protein N termini, and deamidation of Asn and Gln were searched as variable modifications. For a valid protein identification, the following criteria had to be met: at least two peptides detected together with at least seven fragments. The false-positive rate for protein identification was set to 1% based on search of a 5x randomized database, which was generated automatically using PLGS 2.3 by reversing the sequence of each entry. By using the replication rate as a filter, the false-positive rate was further minimized, as false-positive identifications do not tend to replicate across injections due to their random nature. By requiring a protein identification to be made in at least three technical replicates, the effective false-positive rate was lowered to <0.2%. For label-free absolute quantification of protein abundance by mass spectrometry based on the TOP3-method (*Silva et al., 2006*), data were post-processed using ISO-Quant software (*Distler et al., 2014*). The relative abundance of a protein in myelin was accepted as altered if both, significant according to unpaired two-tailed t-test and exceeding a threshold of 25%.

## Immunoblotting

Immunoblotting was performed as described (*Werner et al., 2007*). Antibodies were specific for SEPT2 [ProteinTech Group; 1:500; yielding similar results as a previously described antibody (*Buser et al., 2009*)], SEPT4 (IBL; 1:500), SEPT7 (IBL; 1:5000), SEPT8 (ProteinTech Group; 1:2000), ANLN (Acris AP16165PU-N; 1:1000), MAG (Covance; 1:500), myelin basic protein (MBP; DAKO; 1:500), PLP/DM20 (A431; 1:5000), cyclic nucleotide phosphodiesterase (CNP; Sigma; 1:1000), myelin-oligodendrocyte glycoprotein (MOG; 1:5000; kindly provided by C. Linington, Glasgow), ATPaseα1 (abcam; 1:2500), actin (Milipore; 1:5000), beta3-Tubulin (TUBB3/Tuj1; Sigma; 1:5000), or beta4-Tubulin (TUBB4; Sigma; 1:500). Antibodies specific for phosphorylated or total Akt or Erk1/2 were obtained as a kit and applied as suggested by the manufacturer (CellSignaling, Leiden, The Netherlands). Secondary HRP-coupled anti-mouse, -rabbit, or -goat antibodies were from dianova. Immunoblots were scanned using the Intas ChemoCam system.

## Quantitative RT-PCR

qRT-PCR was essentially as described (*de Monasterio-Schrader et al., 2013*). Briefly, corpus callosi of 10-week-old male mice of the indicated genotypes were homogenized in TRIzol (Invitrogen) using Polytron PT 3100 (Kinematica). RNA was extracted and purified using RNeasy Miniprep kit (Qiagen). The integrity of purified RNA was confirmed using the Agilent RNA 6000 Nano kit and the Agilent 2100 Bioanalyser (Agilent Technologies). cDNA was synthesized using random nonamer primers and the SuperScript III RNA H Reverse Transcriptase (Invitrogen). Quantitative RT-PCR was performed using the Power SYBR Green PCR Master Mix (Applied Biosystems) and 7500 Fast Real-Time PCR system (Applied Biosystems). mRNA abundance was analyzed in relation to the mean of the standards *Ppia* and *Rps13*, which both did not differ between genotypes. Statistical analysis was performed in GraphPad Prism 6.0. Primers were specific for *Sept2* (forward 5'-TCCTGACTGA TCTCTACCCAGAA, reverse 5'-AAGCCTCTAT CTGGACAGTTCTTT), *Sept4* (forward 5'-ACTGACTT-GT ACCGGGATCG, reverse 5'-TCTCCACGGT TTGCATGAT), *Sept7* (forward 5'-AGAGGAAGGC A-GTATCCTTGG, reverse 5'-TTTCAAGTCC TGCATATGTGTTC), *Sept8* (forward 5'-CTGAGCCCCG GAGCCTGT, reverse 5'-CAATCCCAGT TTCGCCCACA), *Anln* (forward 5'-ACAATCCAAG GACAAA-CTTGC, reverse 5'- GCGTTCCAGG AAAGGCTTA, *Ppia* (forward 5'-CACAAACGGT TCCCAGTTTT, reverse 5'-TTCCCAAAGA CCACATGCTT), and *Rps13* (forward 5'-CGAAAGCACC TTGAGAG GAA, reverse 5'-TTCCAATTAG GTGGGAGCAC).

## Nerve conduction velocity measurement

Nerve conduction velocity in the CNS was measured in vivo on 39 adult male mice (23 mice at the age of 6 months: 9 WT, 7 KO, and 7 COND; 16 mice at the age of 18 months: 6 WT, 6 KO, 4 COND). Electrophysiology was essentially as described (*Dibaj et al., 2012*; *Steffens et al., 2012*), with moderately modified surgery procedure. Briefly, anaesthesia was initiated by 80 mg/kg pentobarbital injected i.p. The rectal body temperature was measured and kept at 37°C by a heated plate. After cannulation of the jugular vein, anesthesia was continued with 40–60 mg methohexital per kg and hour. Tracheotomy was performed, and a tube for artificial ventilation was inserted. Active respiratory movements were abolished by paralysis with pancuronium (800 μg per kg supplemented i.p. every hour) and artificial ventilation with a gas mixture of $CO_2$ (2.5%), $O_2$ (47.5%), and $N_2$ (50%) at 120 strokes/min (100–160 μl/stroke depending on the body weight). The vertebral column was rigidly fixed with two custom-made clamps. Electrocardiograms were monitored throughout. Changes of heart rate and temperature were used to control the anaesthetic state. Blood $O_2$ saturation was monitored by a sensor in the inguinal region (MouseOx system, Starr Live Sciences Corp.). Laminectomy was performed from vertebrae TH13 to L5 to expose spinal cord segments L1-L4 and dorsal roots L3-L5. For electrophysiology, the spinal cord was covered with mineral oil. Stimulation and recording were performed with bipolar platinum electrodes. Rectangular constant voltage pulses were used with a duration of 0.1 ms to stimulate dorsal root L4. Supramaximal stimulation strength of 5T was used, which is 5 times the electrical threshold for the lowest threshold fibres (T=threshold; thresholds were frequently tested and adjusted during the experiment). Recording was performed with a sampling rate of 50 kHz on the ipsilateral fasciculus gracilis at spinal cord level L1. The signal was appropriately pre-amplified and filtered. At the end of the experiment, the distance between

the stimulation electrode (cathode) and the recording electrode was measured in situ by using a thin cotton thread.

## Acknowledgements

We thank M Kinoshita, C Linington, K-I Nagata, and W Trimble for antibodies, U Suter for mice, M Schindler for blastocyst injection, A Fahrenholz, R Jung, and T Ruhwedel for technical assistance, J Ficner for help with graphics, ED Schomburg for help with electrophysiology, and C ffrench-Constant, O Jahn, and members of our department for discussions. *Plp*$^{Cre-ERT2}$ mice have been used in collaboration with U Suter. This work was supported by a grant to HBW from the Deutsche Forschungsgemeinschaft (DFG WE 2720/2-1). The authors declare no conflict of interest.

## Additional information

### Funding

| Funder | Grant reference number | Author |
| --- | --- | --- |
| Schweizerischer Nationalfonds zur Förderung der Wissenschaftlichen Forschung | 31003A-125210 | Nicole Schaeren-Wiemers |
| Schweizerischer Nationalfonds zur Förderung der Wissenschaftlichen Forschung | 31003A-141185 | Nicole Schaeren-Wiemers |
| European Research Council | Advanced Grant | Klaus-Armin Nave |
| Deutsche Forschungsgemeinschaft | WE2720/2-1 | Hauke B Werner |

The funders had no role in study design, data collection and interpretation, or the decision to submit the work for publication.

### Author contributions

JP, Performed electron microscopy and all experiments not specified below, Conducted statistical analyses, Designed experiments, Contributed to writing the manuscript, Contributed to revising the manuscript and approved of the version to be published; MSE, Performed electron microscopy (acquisition and analysis of data), Contributed to revising the manuscript and approved of the version to be published; ST, Performed differential proteome analyses (acquisition and analysis of data), Contributed to revising the manuscript and approved of the version to be published; KK, Performed qRT-PCR (acquisition of data), Contributed to revising the manuscript and approved of the version to be published; PD, Conducted electrophysiological measurements (acquisition, analysis and interpretation of data), Contributed to revising the manuscript and approved of the version to be published; WM, Performed electron microscopy (analysis and interpretation of data), Contributed to revising the manuscript and approved of the version to be published; SG, Supplied *Pten*$^{flox/flox}$; *Cnp*$^{Cre/WT}$ mice, Contributed essential reagents, Contributed to revising the manuscript and approved of the version to be published; NS-W, Contributed to analyzing and interpreting data, Contributed to revising the manuscript and approved of the version to be published; K-AN, Contributed to analyzing and interpreting results and to writing the manuscript, Designed experiments, Contributed to revising the manuscript and approved of the version to be published; HBW, Contributed to analyzing and interpreting results, Designed experiments, Conceived the study and wrote the manuscript, Revised the manuscript and approved of the version to be published

### Author ORCIDs

Hauke B Werner, http://orcid.org/0000-0002-7710-5738

### Ethics

Animal experimentation: All experiments were approved by the local Animal Care and Use Committee in agreement with the German Animal Protection Law and by the Niedersächsisches Landesamt

für Verbraucherschutz und Lebensmittelsicherheit (LAVES); permit numbers 33.9-42502-04-11/0418 and -14/1677.

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
