## [Decision Letter]

Thank you for submitting your article "Septin/anillin filaments scaffold CNS myelin to accelerate nerve conduction" for consideration by *eLife*. Your article has been reviewed by two peer reviewers, and the evaluation has been overseen by a Reviewing Editor and Gary Westbrook as the Senior Editor. One of the two reviewers has agreed to reveal his identity: David Lyons (Reviewer #2). The reviewers have discussed the reviews with one another and the Reviewing Editor has drafted this decision to help you prepare a revised submission.

Summary:

This is an interesting and important paper that identifies a novel role of associated septin/anillin filaments in scaffolding the axon/myelin unit in late stages of myelination. Genetic disruption of Sept8 resulted in myelin outfoldings that have been observed in myelin related disorders and aging. The data are compelling and well presented and represent a significant advance for the myelin field and more broadly. The reviewers did not have substantive suggestions for additional experiments. Both raised a few questions and points for clarification that should included in the discussion of the revised manuscript. Once these and minor comments are addressed, this paper will make a valuable contribution to the field.

Reviewer #1:

In this paper the authors perform a proteomic comparison among myelin mutants that have a common phenotype: myelin outfoldings. Amazingly, they identify as a common feature the reduction in septins. They demonstrate this reduction by quantitative mass spectrometry and immunoblotting. They then demonstrate through confocal microscopy and electron microscopy that septins form a filamentous structure that lines the adaxonal membrane of myelin. The septins also form a putative complex with Anillin, and the authors show that this protein colocalizes with septins and is dramatically reduced in the myelin mutants with outfoldings. Consistent with the idea that septin-anillin-phosphoinositide interactions are important, they show that *Pten* cKO mice have dramatic reductions in septins and anillin, and strikingly also have myelin outfoldings. Finally, the authors generate a septin8 conditional knockout and show that by removing septin8 from myelinating oligodendrocytes this results in myelin outfoldings and reduced nerve conduction velocity. These experiments may also explain myelin changes in aging since aged mice also have outfoldings and reduced levels of septins.

Overall I thought the data were excellent and very compelling. I enjoyed reading the paper and think it is a very nice story. I have only one major comment: it is not clear to me why septins form filaments along the axon. Is the idea that the septins provide structural support for adaxonal cytoplasmic channels that might be a 'weak-link' in the structure? If not, why are the septins not evenly distributed around the myelin sheath along the adaxonal membrane? Perhaps this could be clarified. In any case, I think this is a great paper and appropriate for *eLife*.

Reviewer #2:

Patzig et al. submit "Septin/anillin filaments scaffold CNS myelin to accelerate nerve conduction" for consideration at *eLife*. In this paper the authors outline a novel role for septin filament scaffolding in stabilising the growth of myelin sheaths in the central nervous system. The authors show that this is essential for normal myelinated axon function and to prevent the age associated pathology of myelin outfoldings, characteristic of models with disruption to diverse myelin proteins (CNP, PLP, MAG). The quality and presentation of the data are in general outstanding, and I do not have any major concerns with respect to how the data support the claims of the paper. I wonder, however, if the authors could perhaps rework the paper to be of even broader interest to the readership of *eLife*. At the moment it would certainly be of great interest to the myelin community, but in my opinion, the implications of the role of septin function in myelination could be discussed even further.

There is currently great interest in the idea that myelin remains adaptable throughout life, and indeed a paper published recently by some of the authors showed directly that mature myelin sheaths could be stimulated to grow (in thickness) in adulthood. In fact a tamoxifen inducible form of the conditional PTEN knockout used in this study was used to stimulate myelin regrowth. I think it would perhaps be a little unfair to ask the authors to look at how myelin sheath regrowth related to septin deposition, as these are long experiments, but if the authors had the data to hand or wished to do so that would be great. However, I think it would be very useful if the authors could speculate on the role of septin in regulating the ability of mature myelin sheaths to grow. Is it in fact septin deposition by regualting PiP2-PiP3 balance that is the final arbiter of whether a myelin sheath is stable or growing? This would be of great broad interest. Another point that might be of interest for discussion is how septins relate to remyelination: remyelination in the CNS is known to result, at least temporarily in short and thin myelin sheaths- could this be because of premature deposition of septin filaments?

At any rate, these are simply points that could perhaps be made to add a bit more general interest to the paper, and not requests for more experiments. One could dream up many, but this study is excellent in and of itself and well set for publication now.

---

## [Author Response]

*Reviewer #1:*

*In this paper the authors perform a proteomic comparison among myelin mutants that have a common phenotype: myelin outfoldings. Amazingly, they identify as a common feature the reduction in septins. They demonstrate this reduction by quantitative mass spectrometry and immunoblotting. They then demonstrate through confocal microscopy and electron microscopy that septins form a filamentous structure that lines the adaxonal membrane of myelin. The septins also form a putative complex with Anillin, and the authors show that this protein colocalizes with septins and is dramatically reduced in the myelin mutants with outfoldings. Consistent with the idea that septin-anillin-phosphoinositide interactions are important, they show that Pten cKO mice have dramatic reductions in septins and anillin, and strikingly also have myelin outfoldings. Finally, the authors generate a septin8 conditional knockout and show that by removing septin8 from myelinating oligodendrocytes this results in myelin outfoldings and reduced nerve conduction velocity. These experiments may also explain myelin changes in aging since aged mice also have outfoldings and reduced levels of septins.*

Overall I thought the data were excellent and very compelling. I enjoyed reading the paper and think it is a very nice story. I have only one major comment: it is not clear to me why septins form filaments along the axon. Is the idea that the septins provide structural support for adaxonal cytoplasmic channels that might be a 'weak-link' in the structure? If not, why are the septins not evenly distributed around the myelin sheath along the adaxonal membrane? Perhaps this could be clarified. In any case, I think this is a great paper and appropriate for ELife.

We completely agree with the reviewer’s wording of the conclusion that ‘the septins provide structural support ‘in the adaxonal non-compacted myelin layer (‘cytoplasmic channels‘) that ‘might be a 'weak-link' in the structure‘. Indeed, we speculate that an ‘even distribution (of septins) along the myelin sheath ‘may hinder diffusion and transport processes through non-compacted myelin.

In response we have re-checked if this aspect is sufficiently visibly covered in our manuscript. We find that this aspect is already stated visibly, including in the last sentence of our Discussion section. We have thus not changed the manuscript in response to this comment. If the reviewer or editor would further specify what more precisely she/he recommends clarifying here, we will be glad to implement that.

*Reviewer #2:*

*Patzig et al. submit "Septin/anillin filaments scaffold CNS myelin to accelerate nerve conduction" for consideration at eLife. In this paper the authors outline a novel role for septin filament scaffolinding in stabilising the growth of myelin sheaths in the central nervous system. The authors show that this is essential for normal myelinated axon function and to prevent the age associated pathology of myelin outfoldings, characteristic of models with disruption to diverse myelin proteins (CNP, PLP, MAG). The quality and presentation of the data are in general outstanding, and I do not have any major concerns with respect to how the data support the claims of the paper. I wonder, however, if the authors could perhaps rework the paper to be of even broader interest to the readership of eLife. At the moment it would certainly be of great interest to the myelin community, but in my opinion, the implications of the role of septin function in myelination could be discussed even further.*

There is currently great interest in the idea that myelin remains adaptable throughout life, and indeed a paper published recently by some of the authors showed directly that mature myelin sheaths could be stimulated to grow (in thickness) in adulthood. In fact a tamoxifen inducible form of the conditional PTEN knockout used in this study was used to stimulate myelin regrowth. I think it would perhaps be a little unfair to ask the authors to look at how myelin sheath regrowth related to septin deposition, as these are long experiments, but if the authors had the data to hand or wished to do so that would be great. However, I think it would be very useful if the authors could speculate on the role of septin in regulating the ability of mature myelin sheaths to grow. Is it in fact septin deposition by regulating PiP2-PiP3 balance that is the final arbiter of whether a myelin sheath is stable or growing? This would be of great broad interest. Another point that might be of interest for discussion is how septins relate to remyelination: remyelination in the CNS is known to result, at least temporarily in short and thin myelin sheaths- could this be because of premature deposition of septin filaments?

At any rate, these are simply points that could perhaps be made to add a bit more general interest to the paper, and not requests for more experiments. One could dream up many, but this study is excellent in and of itself and well set for publication now.

We agree that it would be interesting to analyze the effect of myelin septins on myelin regrowth (e.g. in Tamoxifen-induced *Pten^flox/flox^**Plp-CreERT2 mice), the membrane’s content of PIP_2_ and PIP_3_, and on remyelination. Some of these aspects we aim to address in follow-up experiments. However, we do not have such data available at this time and thus have not changed our manuscript in response. If requested by the Editor, we could introduce speculative thoughts on these matters into our Discussion section; however considering the absence of experimental data we would rather not.